# Journalistic Quality Criteria under the Magnifying Glass: A Content Analysis of the Winning Stories of World Press Photo Foundation's Digital Storytelling Contest

Rosanna Planer , Alexander Godulla * , Daniel Seibert  and Patrick Pietsch

Institute for Communication and Media Studies, Leipzig University, 04109 Leipzig, Germany
* Correspondence: alexander.godulla@uni-leipzig.de

**Abstract:** This study explores aspects of journalistic quality in complex digital stories. Based on a tailored overview of the potentials of online journalism and digital long form stories for journalistic quality, all available award-winning stories of the subcategory *Interactive* of the *World Press Photo's Digital Storytelling Contest* from 2011 to 2021 (n = 31) are examined according to their structure and journalistic quality criteria using Grounded Theory. The findings add to the long and ongoing research history in journalism and communication studies on the question of what journalistic quality entails and can be used as a basis for further analyses focusing on the technological and structural nature of digital stories and high-quality journalism. The analysis revealed a differentiation between linear stories and chapter stories with linear elements. While a multimedia nature, continuous text and video content prevailed in both forms, they differed in terms of their complexity as well as certain expressions of quality criteria. Gamification and immersion emerged as new yet debatable aspects of journalistic quality in digital stories.

**Keywords:** digital long forms; quality criteria; online journalism; storytelling; World Press Photo Foundation; Storytelling Contest

## 1. Introduction: Achieving Journalistic Quality through Complex Digital Stories?

News outlets turning to digital have encountered many challenges, among them most outstanding the challenge of staying financially stable when content is either free or to be found elsewhere (see Chyi and Ng 2020; Fletcher and Nielsen 2016; Myllylahti 2019; O'Brien et al. 2020). One way of addressing this challenge is to enhance and secure journalism's democratic function through cooperation and demarcation (Esser and Neuberger 2019, p. 196), and thereby generating an added value and quality for the readership.

In this paper, we argue that the complex digital story could be a worthwhile object of researching journalistic quality, for it bears great potential within its characteristics: Complex digital stories became prominent with the much-discussed Pulitzer Prize winning example *Snowfall: The Avalanche at Tunnel Creek*, which appeared in the New York Times in late 2012 as a digital long form (Branch 2012). This journalistic piece is seen as an initiation moment for digital storytelling in online journalism (Godulla and Wolf 2017a), which led to many imitations due to the widespread attention. What distinguishes this story from others is the combination of multiple media types such as text, photos, audios, videos, graphics, and data visualizations (Hiippala 2017, p. 420) as well as high usability and the use of internet-specific potentials. The use of multiple media elements enables a better and more immersive experience of digital stories (Radü 2019, p. 26). Thus, instead of focusing on assessing the stories' overall degree of excellence, we analyze journalistic quality based on the presence and absence of particular form- and structure-related criteria.

This story-oriented form of multimedia Internet journalism has established itself complementing the short and breaking news presentation of topics (Godulla and Wolf 2018). It therefore differs from news journalism in both production and reception speed and mode;

therefore, we argue that the digital long form story can be seen as an innovation field that mirrors current trends in digital journalism, both on the technological and content level, and can become one pillar adding value to digital journalism, communicating quality on a new level. News outlets producing such stories could thereby marry digital innovativeness with high-quality content, inform their readership about highly relevant topics in-depth and thereby play towards their individual preferences in digital news consumption.

To investigate the degree to which news outlets can really achieve this added value for their readership through these kinds of stories, current trends in digital long forms must first be identified and their potentials in terms of journalistic quality need to be assessed. Therefore, we used the *World Press Photo Foundation's Digital Storytelling Contest* as a starting point for an explorative investigation and analyzed the respective entries of the subcategory *Interactive*. While *World Press Photo* is originally and mainly known for professional photojournalism in the form of individual photos, the establishment of the *Storytelling Contest* elevates "the storytelling aspect of an image over all else" and calls upon "journalism's focus on storytelling to carve out a unique space within photography" (Lough 2021, p. 317). These interactive stories go beyond the integration of only photos, and rather focus on the compelling visualization of a whole journalistic story. They have already been awarded as high-quality stories by an independent jury within an international competition with global outreach.

Analyzing these stories in-depth gives insights into (a) the elements and structure of these stories, and therefore (b) into the potential they carry in terms of expressing journalistic quality on the Internet. Previous studies in the field of visual communication have analyzed submitted photos to the *World Press Photo* competition (i.e., Godulla 2009), but no comprehensive empirical study is yet known for having analyzed the entire subcategory *Interactive*, covering a time span from 2011 until 2021. We are aiming at filling this gap, and thereby contributing to the wider discourse about journalistic quality on a global level.

Therefore, the paper pursues the goal of analyzing long form stories and identifying related quality criteria. We thereby aim to answer the following question: *Which different approaches to digital long form stories can be revealed through an in-depth analysis of the Interactive subcategory of the World Press Photo's Digital Storytelling Contest, and which journalistic quality criteria can be identified in these stories?*

## 2. Quality Criteria in (Online) Journalism: A Tailored Overview

### 2.1. From Journalistic Independence to Innovative Interactivity

Quality in journalism has received research attention for decades and fills whole books and journals. Therefore, we do not make a claim of comprehensively reflecting on the topic in this article. It is, however, necessary to refer to the classic and traditional journalistic criteria such as transparency, objectivity, or diversity (Neuberger 2011, p. 71), which shape the journalistic self-image to a great extent. These criteria apply not only to traditional print or broadcast media, but also to digital offerings and are valued by recipients (Neuberger 2012; Urban and Schweiger 2014). Nevertheless, online journalism has brought additional characteristics to this debate. In this context, Meier (2003) presents a canon of ten quality criteria which is divided into two levels:

The first quality at the level of editorial action deals with *independence* and the journalistic *separation norm* (Meier 2003), which is seen as an essential characteristic of journalistic professionalism. According to this, journalists can only fulfill their public task if they work independently of private or business interests of third parties and of personal economic interests (Meier 2003, p. 249). This is especially important in digital journalism. Due to the new technical possibilities, the separation standard is often circumvented.

The *accuracy, originality* and *research quality* of content are also attributed to the first level and are inescapable for the credibility of journalistic information. However, digitization has an ambivalent effect on this aspect of journalistic quality (Lilienthal et al. 2014, p. 32). While fact checking is possible without problems and at any time for all users of the Internet, this simplicity tempts people to use only search engines for research. According

to Meier (2003), this behavior can lead to duplication of content due to repeated use, which ultimately decreases originality, clearing the way for spreading fake news and developing echo chambers. In addition, sources on the Internet can be manipulated more easily and false reports are spread rather simply (Sturm 2013, p.16). Due to this, a competent and extensive verification of information is essential. In the course of the implied fake news debate on the Internet (Tandoc et al. 2018), and the rise of deepfake technology, which poses numerous challenges to media coverage (Godulla et al. 2021), the public in particular is dependent on good research quality (Nuernbergk and Neuberger 2018, p. 104).

*Topicality* describes the temporal dimension (Sturm 2013, p. 17) and is thus an essential element in news journalism (Lilienthal et al. 2014). In this context, online media have the advantage that already published content can be constantly updated, and that news tickers can provide the public with information quickly. Within long form stories specifically, topicality or newness of the content is secondary, since the comprehensive and partly even investigative nature of a topic are most important.

At this point, the aspect of *relevance* as a journalistic quality criterion comes into play. For complex digital stories, relevance legitimizes the expense of long production cycles and many required resources. In digital journalism, there is a shift in relevance due to the tabloidization of journalism and through the increasing work with algorithms (Lilienthal et al. 2014, p. 34). Related to that, Lilienthal et al. (2014, p. 39) add *discoverability* as another extended quality in this context. As algorithms determine the content displayed and more and more people use search engines, the published information must also be detectable for users.

Another journalistic quality that can be shaped within the newsroom is *interactivity*. What in traditional journalism was the ability of an editorial team to engage in dialogue (Meier 2003, p. 255) is somewhat different on the Internet: This term refers to the integration of users into the website, which can be based on the user's contribution, for example through blogging (Steensen 2013), commenting (Domingo et al. 2008), participating in reader polls (Peters and Witschge 2015), as well as based on the exchange between user and producer through live chats (Sundar et al. 2014) or a dialogue on the editorial content (Peters and Witschge 2015). Further dimensions of interactivity in digital journalism also exist in terms of a glitch-free consumption (Jacobson et al. 2015), the ease of adding information and the complexity of choice (Massey and Levy 1999), as well as in a time-related dimension of real-time participation (Sundar et al. 2014) or simultaneous contribution (Appelgren 2017). Digital long form stories could make use of a great potential in terms of interactivity.

### 2.2. From Elaborate Crossmediality to Sophisticated Computerization

With *crossmediality*, Meier (2003) proposes a new quality criterion, which is based on cross-media working methods in an editorial department. The author thus concludes the first editorial level of qualities and turns to the second, which focuses on *product-related qualities*. The first mentioned aspect is the diversity of perspectives and the information content. Both are necessary to be able to speak of a democratic media system (Lilienthal et al. 2014, p. 33). Non-linear stories offer more narrative possibilities and provide for a more complex presentation of content by illuminating a wide variety of topics in parallel (Meier 2003). Hence, digital long forms can contribute to a democratic discourse in that they can make new voices heard and take up the space to present different perspectives on a topic.

Next to crossmediality, the product-related qualities also consist of *comprehensibility* and *usability*. Content does not add value to users if it is not formulated in an understandable way. This applies to both traditional media and digital journalism offerings. Based on Nielsen's (1990) multifaceted examination of usability, Meier (2003, p. 259) sees the term as an extension of comprehensibility and defines it in terms of the utility value of an offering. In this sense, it refers to the optimization of the website and the associated improvement of navigation as well as the general user-friendly structuring of the content on it. Accordingly,

usability expresses how well an online product can be used and accessed in a specific context to achieve specific goals (Hooffacker 2020, p. 87).

The task of the quality criteria *excitement, sensuality* and *vividness* is to enliven the reception and increase the absorption of information (Meier 2003). In online journalism, this quality finds great appeal due to the continuous development and proliferation of multimedia elements. Combining them provides visual variety and conveys content to recipients in a more vivid way. *Transparency* refers to the disclosure of sources (Meier 2003, p. 261). What often leads to space problems in print media can be solved in digital media with external links. Users can thus trace the origin of the train of thought with a single click. However, despite the already common practice, according to Godulla and Wolf (2017b), there is a risk that users will distance themselves from the reception process through external forwarding.

Finally, *mechanization, automation, and computerization* evoke entirely new potentials and specializations (Lilienthal et al. 2014). Godulla and Wolf (2017b) therefore assume that, in addition to the classic quality criteria, technical-formal properties also play an essential role in the evaluation of online offerings, which come into play for digital long forms, too.

### 3. Digital Long Form Stories: About Cognitive Containers and Digital Innovativeness

Digital storytelling in journalism combines the potentials of media elements such as text, photography, graphics, video, animations, and audio to prepare non-fictional content in an immersive manner (Godulla and Wolf 2018). This results in what Dowling and Vogan (2015) refer to as "cognitive container", in which elements are blended into "a coherent whole in its own self-contained package [that] carries the benefit of shielding the reader from the distractions of the open web" (Dowling 2019, p. 31). In this respect, elaborate designs, the extensive integration of multimedia elements, as well as the communication of background information speak against an actuality-centered journalism and rather circle around in-depth topics such as environmental issues or political conflicts (Planer and Godulla 2020); at the same time, the format in itself conveys that the audience should bring some time with them when consuming a story (Planer and Godulla 2020), showing parallels to the traditional magazine journalism. Furthermore, the story itself can be part of crossmedia or transmedia structures (Gambarato 2016) since the Internet as the basis for this journalistic format enables comprehensive narratives due to the technical possibilities.

On the one hand, the combination of multimedia elements and the depth of storytelling create new reception experiences for users. On the other hand, digital storytelling also opens new possibilities for producers to prepare journalistic content. For photojournalists, for example, digital stories help maintain their professional integrity through "telling a different side of a story", "inspiring others and invoking change", as well as "memory-keeping" (Mortensen and Gade 2018, p. 582). Planer et al. (2020) argue that these mentioned potentials are normative arguments in favor of complex digital storytelling, especially when it comes to digital long forms which can be of several different structures as outlined in the following.

*Classifications of Digital Stories and Their Inherent Structures*

Longhi and Winques (2015) conclude that the structure of a digital story is guided by two narrative forms, which they refer to as vertical and horizontal. In vertical narratives, reception occurs through what is called scrolling (Longhi and Winques 2015). Integrated elements appear as users move through the story by scrolling up and down, suggesting a linear long form structure. In horizontal storytelling, chapters or sections form the basis of the story, which initially suggests a non-linear structure and encourages users to use it independently. However, once a chapter is selected, reception again follows the principle of vertical storytelling (Longhi and Winques 2015, p. 116).

Hernandez and Rue (2015) also structure digital long forms into their continuous, comprehensive, and immersive nature (Hernandez and Rue 2015, p. 98). First, continuous stories are characterized by a classic linear structure (beginning, middle, end) and are also

tied to a primary medium (Hernandez and Rue 2015, p. 98). Due to these characteristics, continuous stories are also attributed a so-called lean-back character, which is expressed primarily through reading or watching on the side of the users (Hernandez and Rue 2015, p. 98). Second, the narrative form of comprehensive stories is based on a rather free reception of the story. The lack of linearity and the variability of the elements ensures interest-based use by the recipients. This is made possible by the subdivision into sections, in which additional individual narrative forms can emerge (Hernandez and Rue 2015, p. 101). The extensive structure and informative character are therefore particularly suitable for addressing multi-layered topics. Third, the immersive narrative form in contrast shows a clear tendency towards interactive elements and gaming structures. Various audiovisual media forms, such as interactive videos, animations, graphics, or background music and the frequent use of full screen mode are intended to ensure that users are fully immersed in the digital long form (Hernandez and Rue 2015, p. 105).

Godulla and Wolf (2015a, 2015b, 2017b, 2018), on the other hand, distinguish digital long forms according to their structure and effect, which shows some parallels and overlaps with Longhi and Winques (2015) as well as with Hernandez and Rue (2015). The authors fundamentally differentiate between the formats of scrollytelling, web documentations, and selective multimedia stories. The basic framework of a scrollytelling story is linear in orientation (Godulla and Wolf 2017b) and does not necessarily have to be text-based but can also function as an image-centered story (Godulla and Wolf 2017b). Furthermore, an essential feature in the alignment is the integration of multimedia elements. Whether these elements are used by their readers is up to them and dependent on their interest. However, this diversity provides depth to the story and, to a certain extent, decelerates the reception.

The web documentary or interactive (web) documentary (Ducasse et al. 2020; Pavlic and Pavlic 2017), on the other hand, is characterized by opulent visuals and the elaborate use of moving video or even animated images (Sturm 2013, p. 118). With the help of this non-linear structure, users can make their own way—independent of the dramaturgy—through the content, leaving traces behind (Mundhenke 2016). Due to the diverse integration of a wide range of elements and the elaborate production, the web documentary is also described as the supreme discipline (Sturm 2013, p. 118), being connected to the original documentary genre.

The third form of digital long forms specializes primarily on mobile devices and is prepared mainly for the reception on tablets. Users can navigate through a basic linear structure pre-planned by the producers using various touch, swipe or tap gestures (Godulla and Wolf 2017b). In the selective multimedia story, the contributions are oriented horizontally, but this does not exclude branching off through multimedia elements such as audio slideshows, 360-degree panoramic photos, read-aloud functions, or hotspots (Godulla and Wolf 2015a).

On the basis of the outlined various structures of complex digital stories, we argue that these stories have the potential to carry a significant added value for digital journalism in a networked society, making use of the development that "new digital journalism has sparked a renaissance in deep reading and viewing" (Dowling 2019, p. 1).

## 4. Potentials for Journalistic Quality in Digital Long Forms

### 4.1. Usability and Multimedia Elements: The Necessity of Their Flawless Interconnection

The criteria mentioned in Chapter 2 can be applied to digital long forms, too, since they are a distinctive format within digital journalism. Compared to digital news stories, however, long form stories require not only more time, but also more human resources due to the different narrative forms, hypertext, multimedia elements, or even error-free navigation (Godulla and Wolf 2015a). This circumstance eventually leads to journalism overlapping with parts of computer science and graphic design (Godulla and Wolf 2017b).

The composition of a comprehensible navigation, which leads the user through the story without orientation problems, also falls into that area of responsibility. Accordingly, *usability* is a criterion which, based on online journalism and Meier's (2003) explanations,

also plays a decisive role in digital long forms and was therefore used as one of several quality criteria for our analysis. Current research on usability focuses, among other things, on the question of which navigational elements (e.g., chapters, progress bars, connection of buttons and scroll bars) are appropriate for reception and for an overview of the story. Most stories use a clickable menu that makes it easier for users to select and navigate content (Geis 2014; McKenna et al. 2017; Pavlic and Pavlic 2017; Pope 2020). Consequently, the term usability is understood as navigation within the long forms, which is afflicted with the task of simplifying orientation within the story by means of navigational elements that draw the user's attention to special features, thus enabling intuitive usability.

Against the thematized technical background, however, usability as a sole feature is too little. Since digital long forms move within the context of online journalism and are also tangential to mobile offerings, Godulla and Wolf (2017b) assume that technical-formal characteristics also play a role in the assessment of digital journalistic offerings. In this regard, in addition to usability, the quality criteria *utility, multimedia, interactivity, participation, selectivity* and *linking* are added (Godulla and Wolf 2017b, p. 31), which are presented in the following due to their high relevance.

*Utility* is defined by the problem-free functioning of a contribution and thus, according to the authors, also includes the resolution of photographic and video content. Studies on the quality criterion of utility focus for example on the functioning of posts and analyze loading processes of pages and of content (Godulla and Wolf 2017b). The term *multimedia* has received considerable research attention (i.e., Deuze 2004; Jacobson 2012; Meier 2002; Song 2018), with scholars pointing out the implementation of multiple media formats into a journalistic offering, such as written word, photography, video, audio, photo essays, graphics, animations, and the like (Deuze 2004; Kartveit 2017; Jacobson 2012; Wolf 2014). It is additionally supplemented by the integration of new forms of presentation such as 360° photography or audio slideshows (Godulla and Wolf 2017a; Song 2018). Next to the multitude of media elements, the accessibility on multiple platforms (Song 2018) and the nature of the multimedia flow (Pincus et al. 2016) complement the scholarly discussion around the term. Furthermore, multimedia elements complement the narrative (Giles and Hitch 2017) and provide a means of creating synergy to deepen content. However, their number should be carefully considered to avoid audiovisual overload for recipients (Godulla and Wolf 2017b; Hiippala 2017; Lassila-Merisalo 2014; Pincus et al. 2016; Tulloch and Ramon 2017).

### 4.2. Sharing Is Caring: About so Far Underestimated Factors of Participation, Transitivity & Co

Multimedia and *interactivity* are interconnected, since, to encourage audience interactivity, producers draw on multimedia and integrate special graphics or animated gifs into their posts to attract new users by sharing, discussing, and continuing the story on social media, thus increasing click-throughs (Lassila-Merisalo 2014; Riskos et al. 2019; Weber et al. 2018). When it comes to interactivity, which has been referred to above, the focus in long form stories seems to lie even more on the option of sharing and forwarding in the context of social media or via E-mail (Godulla and Wolf 2017b).

The notion of *participation* bears overlaps with interactivity, but also bears some further distinctions: Participation can either happen *in* news or *through* news, the former referring to participating in a public debate on the internet, the latter referring to co-deciding on editorial content on a professional and managerial level (Peters and Witschge 2015; Spyridou 2019). Hence, participation can occur through interactive tools (Peters and Witschge 2015); however, the extent to which participation is allowed depends on the producers and the technical affordances.

In an overarching sense, *selectivity* is responsible for the structure of a digital long form. In order for the elements to really create a seamless whole or the so-called cognitive container (Dowling and Vogan 2015), they are combined in sense units, or segments, and united in a narrative structure (Godulla and Wolf 2017b). Selectivity can be subdivided into the linear and the non-linear narrative form, the former being characterized by the features of a continuous vertical—rarely also horizontal—story and a primary medium

with a lean-back character (Hernandez and Rue 2015). The latter, for example, includes the concentric form and game format narrative. Non-linear narrative structures in the context of selectivity may have a beneficial effect on users by providing helpful choices for navigation (Pope 2020) and by providing the opportunity for users to receive the long form according to their interests and at a depth of their own choosing (Tulloch and Ramon 2017; Vázquez-Herrero 2021).

*Linking* is partly related to source transparency, especially when it comes to external links. Apart from that, internal and cross-media linking also count to this internet-specific quality. Internal linking, on the one hand, is established for various offerings in order to control the receptive flow and keep users in their own cosmos (De Maeyer 2019). Cross-media linking, on the other hand, serves to draw attention to further online as well as offline services (Godulla and Wolf 2017b). Taken together, the criteria presented by Meier (2003), Godulla and Wolf (2017b), and enriched by further scholars, form the basis for evaluating Internet-specific quality. In this context, the criteria serve as an assistance for editors in the creation and evaluation of digital long forms (Schumacher 2009) and should additionally encourage recipients to use the offering on a sustained basis (Godulla and Wolf 2017b).

With regard to a longer and sustained reception, *transitivity* can furthermore be added. This rather technical element of a long form story is responsible for maintaining the tension and elegantly transitioning from one medium to another (Radü 2019, p. 245). When compared to the back-and-forth nature of the hypertext principle, transitivity seems rather contradictory, stringently merging elements, but transitions have been proven to pay off especially with many different media forms to hold the story together (Radü 2019, p. 245). Usability, utility, multimedia, interactivity, participation, selectivity, linking, and transitivity ultimately form the normative and Internet-specific evaluation approach for this paper in order to classify digital long forms in terms of their journalistic quality. Table 1 below shows an overview of the quality criteria for online journalism and digital long forms.

**Table 1.** Overview of identified quality criteria of online journalism and digital long forms.

| Quality Criteria | |
| --- | --- |
| Quality criteria in online journalism | Independence and separation norm (e.g., Meier 2003) |
| | Accuracy, originality and research quality (e.g., Lilienthal et al. 2014) |
| | Topicality (e.g., Sturm 2013) |
| | Relevance (e.g., Meier 2003) |
| | Discoverability (e.g., Lilienthal et al. 2014) |
| | Interactivity (e.g., Domingo et al. 2008; Meier 2003; Steensen 2013) |
| | Crossmediality (e.g., Meier 2003) |
| | Comprehensibility and usability (e.g., Meier 2003; Nielsen 1990) |
| | Excitement, sensuality, and vividness (e.g., Meier 2003) |
| | Transparency (e.g., Meier 2003) |
| | Mechanization, automation and computerization (e.g., Lilienthal et al. 2014) |
| Additional quality criteria in digital long forms | Utility and multimedia (e.g., Deuze 2004; Godulla and Wolf 2017b; Meier 2002) |
| | Participation (e.g., Peters and Witschge 2015; Spyridou 2019) |
| | Selectivity (e.g., Godulla and Wolf 2017b; Hernandez and Rue 2015) |
| | Linking (e.g., De Maeyer 2019; Godulla and Wolf 2017b; Schumacher 2009) |
| | Transitivity (e.g., Hiippala 2017; Jacobson et al. 2015; Radü 2019) |

Having laid out both the characteristics of the format of digital long form stories and the potentials for journalistic quality they are able to carry, we propose the following research-guiding question for our study:

RQ: *Which different approaches to digital long form stories can be revealed through an in-depth analysis of the Interactive subcategory of the World Press Photo's Digital Storytelling Contest, and which journalistic quality criteria can be identified in these stories?*

In order to investigate this question, the World Press Photo Foundation and their Storytelling Contest will be introduced in the following to give an overview of the research object at hand.

## 5. The World Press Photo Foundation and its Storytelling Contest

In April of each year, the jury of the *World Press Photo Foundation* awards the winners of the Digital Storytelling Contest. First launched under the name "Multimedia Contest" in 2011, the contest has since aimed to promote diverse forms of visual storytelling (World Press Photo Foundation 2021b). Like the *World Press Photo Contest*, the *Digital Storytelling Contest* started small: With 42 nominations, the number grew to a substantial 300 submissions in 2020, and 287 productions in 2021 (World Press Photo Foundation 2021b). The stories are judged by an independent jury consisting of digital storytellers and multimedia editors from around the world, who must be active in this field as professional visual journalists.

Producers' submissions should follow the given framework of the contest and have a focus on "[ . . . ] short documentary film and interactive productions [ . . . ] and visual storytelling enabled by digital technologies" (World Press Photo Foundation 2021c). Moreover, in order to adapt to technological and innovative standards, since 2019, instead of one, two highly endowed prizes are awarded: the *World Press Photo Online Video of the Year* and the *World Press Photo Interactive of the Year* (World Press Photo Foundation 2021c). In total, the competition consists of three subcategories: *Short, Long,* and *Interactive*. Both the former and the latter are based on documentary-style videos and feature linear storylines. In addition, they should be produced for the web (World Press Photo Foundation 2021a).

In the *Interactive* subcategory, the name already reveals what is required of the producers: Interactive visual stories whose design is intended to evoke an immersive and/or innovative experience in the recipient (World Press Photo Foundation 2021a). The prerequisite in this context is the combination of video and/or photographic content with animation, graphics, illustration, sound or text (World Press Photo Foundation 2021a).

The conditions of the third subcategory show a clear relation to the characteristics of digital long forms. The multimedia features of the award-winning stories provide a clear parallel to the elaborated quality features. Since only individual case studies or specific explanations (e.g., Hernandez and Rue 2015; Mundhenke 2016; Schlichting 2015; Witte 2014) but no overarching research is available specifically on long forms in this subcategory, we pursue the goal of closing this research gap. Consequently, the subcategory *Interactive* will be the focus of further empirical investigations, in order to draw conclusions towards quality in digital journalism.

Based on the competition website, the selected research object comprises a total of 37 award-winning digital long forms over the period from 2011 to 2021. However, upon reviewing the material, two stories were no longer to be found at the link provided, or the Internet address was assigned to a different host. The abolition of the Flash player in January 2021 also had an impact on the number of stories. In this respect, three contents were no longer playable even before the material was processed. Therefore, a total of 31 stories of the subcategory Interactive from 2013 to 2021 remained for the further empirical investigation, which can be viewed in Table 2.

**Table 2.** Overview of the story sample.

| Year | Name of the Long Form | Production | Origin |
|---|---|---|---|
| 2021 | Reconstructing Seven Days of Protests in Minneapolis After George Floyd's Death | The Washington Post | USA |
| 2021 | Birth in the 21st Century | Cooperative Hat, Lab RTVE, À Punt Mèdia | Spain |
| 2021 | Ukraine: Grey Zone | Lithuanian Radio and Television (LRT) | Lithuania |
| 2020 | Battleground PolyU | China Daily DOCS (Youtube) | UK |
| 2020 | River of Forgiveness | Helios Design Labs | Canada |
| 2019 | Notes From Aleppo | Paradox | Syrian |
| 2019 | Flint is a place | Zackary Canepari | USA |
| 2019 | The Last Generation | Frontline/The GroundTruth Project | USA |
| 2018 | Under a Cracked Sky | The New York Times | USA |
| 2018 | Sin Luz: Life Without Power | The Washington Post | USA |
| 2018 | How 655000 Rohingya Muslims Escaped | The New York Times | USA |
| 2018 | Finding Home | TIME | USA |
| 2018 | From Janet with Love | National Film Board of Canada | Canada |
| 2018 | There once lived . . . | Such Dela | Russia |
| 2017 | The Dig | The Skin Deep/Murmur | USA |
| 2017 | The Fine Line: Simone Biles Gymnastics | The New York Times | USA |
| 2017 | The Injustice System | The Guardian US | USA |
| 2017 | A New Age of Walls | The Washington Post | USA |
| 2017 | The Waypoint | The Washington Post | USA |
| 2017 | Future Cities | Yvonne Brandwjik, Stephanie Bakker | The Netherlands |
| 2016 | Desperate Crossing | The New York Times Magazine | USA |
| 2016 | Life After Death | NPR | USA |
| 2016 | Welcome to Parkersburg West Virginia | Huffington Post | USA |
| 2016 | The Displaced | The New York Times Magazine | USA |
| 2016 | Greenland is Melting Away | The New York Times | USA |
| 2016 | Graphic Memories: Tales From Uganda's Female Former Child Soldiers | European Journalism Center | The Netherlands |
| 2015 | {The And} | The Skin Deep in collaboration with Deep Focus and Topaz Adizes | USA |
| 2014 | A Short History of the Highrise | The New York Times | USA/Canada |
| 2014 | NSA Files: Decoded | The Guardian US | USA |
| 2014 | Hollow | Requisite Media | USA |
| 2013 | Bear71 | National Film Board of Canada | Canada |

## 6. Methodology: Using Grounded Theory to Investigate Complex Stories

In order to evaluate the digital long forms, Grounded Theory (Strauss and Glaser 1980) and the associated problem-solving research action as well as process orientation (Strübing 2018) were applied to continuously revise and concretize the category system. Grounded theory is understood as an interpretative social scientific approach of working and understanding (Pentzold et al. 2018). The basic procedure for generating a theory refers to the constant and repetitive comparison of data according to conceptual similarities and differences (Pentzold et al. 2018). In this context, a central approach is not to get too

hung up on already developed categories and concepts during the evaluation process and to remain open-minded towards the research material.

The evaluation of the material builds on three coding methods that interact according to the parallel working method. First, open coding (related to individual interviews, here: long forms) is carried out, in which access to the material is gained, relevant pieces of material are selected and then coded in detail (Strübing 2018) in order to question pre-assumptions and to identify and conceptualize phenomena in the research material (Strauss and Corbin 1996). In the second step, defined as axial coding (referring to all interviews, here: long forms), formed categories are related to each other (Krotz 2018) and causes, circumstances, and consequences of these different variations are explored (Strübing 2018). Last, selective coding (related to all interviews, here: long forms) takes place, in which the codes are summarized and hierarchized (Krotz 2018). After selecting a core category, complementary categories are connected and related to each other around it (Strauss and Corbin 1996).

In the first step of the evaluation process, the long forms were compiled into documents in the form of screenshots and made available as a PDF file. This was since video and animated content in particular did not meet the usual requirements for evaluation during a test run. In the second step, the open coding of the stories was carried out in the period from 21 June 2021 to 7 July 2021, resulting in a preliminary category system. To code in a more goal-oriented manner and to avoid possible inaccuracies, the codes were given coding instructions and descriptions using the memo function with the help of the analysis software MAXQDA. This process was accompanied by axial and selective coding. For example, in the context of usability, the subcategories "active and passive navigation and operating instructions" resulted, which in turn united certain characteristics among themselves. A total of 10,742 codes were collected in the course of the coding process.

With regard to the further procedure, the theoretical sampling was based on Strübing (2018) and the accompanying minimum and maximum contrasting. Starting with the first long form to be studied, which was based on a similarity analysis, a single case study was finally conducted. Thereupon, further stories were added, which were similar to each other in various characteristics. Since two basic distinctions in the long forms have already been established from the triage and screen plotting, minimal contrasting was performed until automatically only stories with other features remained that either confirmed or rejected the previous theory and consequently constituted a sub-theory of their own. At this point, maximum contrasting began.

In addition to the similarity analysis, the Code Matrix Browser and the Code Relations Browser of MAXQDA were used. MAXMaps, which presents the relationships and differences between documents and codes on a map, provided visual support for the evaluation. Based on the identified quality criteria of online journalism and digital long form stories and building on the open, axial and selective coding processes, the category system shown in Table 3 was created. Due to the partly large number of sub-sub-categories, only frequently coded categories are listed for the respective main categories.

**Table 3.** Main and subcategories of digital long form stories derived through Grounded Theory.

| Category | Sub-Category | Sub-Sub-Category |
|---|---|---|
| Selectivity | Linear<br>Chapter<br>Non-linear | - |
| Multimediality | Text<br>Photo<br>Video<br>Audio<br>Visualization | (e.g., continuous text, textbox quote)<br>(e.g., full screen, single photo, background photo)<br>(e.g., full screen video, video loop with/ without audio)<br>(e.g., single audio, background sounds, music)<br>(e.g., simple, selective, complex data visualization) |
| Usability and Navigation | Active user assistance<br>Passive user assistance | (e.g., buttons, timestamps, chapter overview)<br>(e.g., navigation and reception note, progress bar, info point) |
| Transitivity | Click<br>Scroll<br>Automatic Transition | (e.g., parallax scrolling) |
| Linking | Hyperlink (intern)<br>External link<br>Cross medial link | (e.g., external document, advertising) |
| Interactivity | Within the story<br>Whole story | (e.g., Twitter, Permalink, Facebook)<br>(e.g., e-mail, Facebook, Twitter) |
| Participation | User-generated content<br>Survey<br>Check list<br>Rating<br>Comment function<br>Contact form | - |
| Utility | Progress bar<br>Bugs | (e.g., not displaying image, not functioning audio/link) |
| Gamification | Open-world environment<br>Click and drag transformation<br>Roll over<br>Puzzle<br>Multiplayer game<br>Jump and run | - |

## 7. Results: Towards a Detailed Differentiation of Digital Long Form Stories

*7.1. Case Contrasting: Linear Stories, Chapter Stories, and Non-linear Stories*

The choice of the first long form was preceded by various considerations and observations. Based on the process of screenshotting and coding all stories, it is noticeable that in the existing long forms, two characteristics or subcategories of selectivity are primarily decisive for the narrative structuring: the linear structure and the chapter structure, which together made for the majority of the stories with two exceptions, which were classified as non-linear structure. In addition, chapters were also found to have a predominantly linear structure (see Table 4).

Due to these two strong characteristics, two partial theories are basically aimed at, which are subsequently compared. Due to the small number of non-linear stories identified, primarily linear and chapter stories will be focused on in the following.

**Table 4.** Overview of long forms by type of selectivity.

| No. | Linear | Chapter | Non-Linear |
|-----|--------|---------|------------|
| 1 | Reconstructing Seven Days of Protests in Minneapolis After George Floyd's Death | Birth in the 21st century | {The And} |
| 2 | Battleground PolyU | Ukraine: Grey Zone | Under a Cracked Sky |
| 3 | Sin Luz: Life Without Power | River of Forgiveness | |
| 4 | How 655000 Rohingya Muslims Escaped | Notes From Aleppo | |
| 5 | Finding Home | Flint is a place | |
| 6 | From Janet with Love | The Last Generation | |
| 7 | The Fine Line: Simone Biles Gymnastics | There once lived . . . | |
| 8 | The Waypoint | The Dig | |
| 9 | Desperate Crossing | The Injustice System | |
| 10 | Life After Death | A New Age of Walls | |
| 11 | Greenland is Melting Away | Future Cities | |
| 12 | Graphic Memories: Tales From Uganda's Female Former Child Soldiers | Welcome to Parkersburg West Virginia | |
| 13 | | The Displaced | |
| 14 | | NSA Files: Decoded | |
| 15 | | A Short History of the Highrise | |
| 16 | | Hollow | |
| 17 | | Bear71 | |

*7.2. Linear Stories: Long Texts, Full-Screen Visualizations, and Low Utility and Participation*

In linear stories, continuous text proved to be the most stringent feature, increasingly developing into a constant companion of linearity. In addition, text boxes represent a new overarching text form. The multimedia element full-screen photo was newly included in the category system during the open coding process and represents one of the most frequently used features in the stories. The element video also received more diverse distinctions during coding; here, the full-screen video appeared fairly often. In general, full-screen formats in the form of photos or videos as well as video loops make a considerable contribution to the visual support of the story. Furthermore, auditory overlaps support the visual components by appropriate sounds or music from the background. As in the case of continuous text, the versatile use also offers room for errors. Complex data visualizations are used frequently, first and foremost in the form of maps. The further story elements are mainly connected by scrolling. The producers integrate progress bars for better orientation, link across different media and, especially at the beginning of the long form, process various social media links to share the whole story.

On the other hand, there are still many differences within the different linear long forms, which have to be accepted as partial theories for the time being and are a first indication for the individuality as well as for non-standardized realizations in digital long forms. When it comes to the quality criteria, usability is realized through progress bars, and transition or transitivity is realized through scroll transitions as a common feature in linear long forms. The quality criterion linking is especially pronounced in the cross-media form and proves to be a constant feature in the majority of the linear stories. In terms of interactivity, the option to share the long form as a whole proved to be common.

Facebook and Twitter were evidenced in this regard, while the email sharing option was also common. The different coding runs as well as the comparison also brought to light that both participation and utility were very weak, but generally, the linear long forms proved to be relatively error-free.

*7.3. Chapter Structure: More Complexity and Finer Nuances*

When examining long forms with chapter structure, the first thing that stands out is that the number of chapters varies greatly, ranging from a minimum of two to a maximum of twelve chapters. The most constant element, as in linear stories, is the continuous text. A parallel is also the versatile overlap with other elements. Thus, in the comparison between linear and chapter, there are no differences purely from the text elements. In both cases, the continuous text is an all-encompassing feature, text boxes and quotations are occasionally added by the producers.

The subcategory photo turned out to be much more comprehensive in the context of the chapter structure. In contrast to linear stories, which are limited to fullscreen, single and background photos, additional features were found in the chapters with screenshot, photo gallery, photo collection, image detail, selective photos and the 360° photo. Furthermore, newly added were the photographic element screenshot, the photo gallery and the photo collection. Compared to linear long forms, only the video gallery could be included in the category system as a new videographic feature. Simple data visualizations were also familiar from linear long forms in the style of line charts. Stories with chapter structure, however, proved to be more complex in this respect, too. Simple data visualizations in most cases stood on their own and were not combined with other elements. The comparison between linear and chapter structure further reveals significant differences in the audio subcategory. In contrast to the low use of background music and background noises in the context of linear long forms, these are integrated significantly more often in the chapters.

The evaluation of the linear long forms already showed how extensively the multimedia elements are combined with features of other quality criteria, which could be observed in the chapters structure as well. However, they differ by an even greater variety of integrated elements, which is consequently also reflected in even more extensive connections among each other. There is a higher number of usability elements found in chapter stories than in linear stories, represented through chapter overviews, navigation elements and operating aids. In this regard, about twice as many hints are inserted in chapter structures. There are also some changes in transitivity in comparison between linear structures and chapters. The original dominance of scroll transitions is far less pronounced in the chapter structure, where clicking is an often-installed transition, too.

When it comes to interactivity, not only the entire story could be shared, as it was the case in linear stories, but also certain parts of the story could be shared. In both cases, Facebook and Twitter dominated as sharing platforms. Which elements could be shared by users in social media thus depends on the given possibilities of the producers. The same is true for the degree of participation. Furthermore, the subcategory gamification—expressed through multiplayer options or drag and drop-feature—was added as a completely new quality dimension for chapter stories, not occurring in linear stories at all. Looking at the quality criterion of utility, few inconsistencies were found in linear long forms. The same can be observed in chapter structure. Missing content and broken links that showed little overlap with the continuous text were the most common errors.

At this point, various aspects can be summarized with regard to the evaluation of linear long forms and those with chapter structure: The chapter stories stood out through a more comprehensive multimedia structure and a higher degree of usability. In addition to the possibility of sharing the entire story in the social networks, interactive sub-options were recorded within the chapter structures that are related to multimedia elements. This possibility is completely denied to linear long forms. Participation as well as utility were rather low in both narrative structures. Gamification is again new in the chapters, but it is only temporarily distributed in individual stories. To complete the analysis, the two

analyzed non-linear long forms stood out from the rest in terms of their narrative form, since they contained less text and were rather heavy in video and a concentric structure, but they nevertheless joined the ranks of the 31 long forms on the basis of overarching characteristics, specifically shown by the emphasis on videographic content.

### 8. Discussion: An Underestimated Format with Still Expandable Potentials

First of all, the results show that the analyzed long forms are much more extensive than assumed in the state of research, technically rendering them even more potential in terms of journalistic quality. Some of the elements emerging through the analysis developed as overarching characteristics of different stories. They then formed overlaps with further quality criteria, which emerged as partial theories. Multimediality turned out to be the most extensive and central criterion of the digital long forms examined here. This can be derived comprehensively in all three narrative forms. The fact that "multimedia is at the heart of its narrative structure" (Hiippala 2017, p. 421) can thus be confirmed. Further outstanding was the extensive use of continuous text in both linear and chapter stories. This seems logical since "what remains to be conquered is the dramatic narrative" which lets "characters go their own way" (Ryan 2009, p. 57). Especially when compared to conventional news stories, interactive digital narratives provide more flexibility and the inclusion of multiple perspectives (Murray 2018, p. 14), "so that we can zoom in and out through time and space and abstraction layers, and across points of view and frameworks of interpretation" (Murray 2018, p. 13).

Another striking characteristic of long form stories was the deliberate use of videographic elements such as full-screen videos, video loops without sound, and videos with sound, which are often made available to users as visual components.

To a much lesser extent, however, data visualizations made an appearance, thus opposing the initial approaches to data journalism. The evaluation of the audio subcategory further showed not only that the defined elements from the research state were not sufficient, but additionally clarified the special use of background audio in the form of sounds, music, and narration. The expressions of the usability quality criterion register a clear tendency towards active navigation and operating aids. In general, however, it was found that a more pronounced usability prevails in the studied chapters than in linear long forms. This uneven ratio can be explained by the more extensive narrative structure and the associated multimedia diversity. Generally, a great importance was attributed to orientation in the stories, which again confirms Dowling and Vogan's (2015) expression of the stories being "cognitive containers".

The most self-contained digital long forms, whose linking structure is concentrated on specific areas in the story, can be used not only to attract attention, but also to provide the simplest possible structure in the long forms, which confirms schema theory, in which schemas serve to reduce complexity (Schmidt and Weischenberg 1994) and provide users with an orientation for proper usability in the form of patterns (Neuberger 2005).

### 8.1. Multimedia Trends and Developments in Digital Long Forms

As the results and the quality criteria show, digital long forms are complex and extensive stories that cannot necessarily be pigeonholed into a standardized category. Rather, individual features evolve, new ones are added, and some fall away over time.

One first observed development lies in the use of lean-back elements. The overarching use of video and photographic as well as textual features contributes to this insight. From those elements, a reading or watching consumption of the story can be derived, which is usually perceived as more intense, profound, and narrative (Hernandez and Rue 2015, p. 170). Although the setting in the stories differs throughout, again arguing for diversity in the long forms and against a standardized how-to guide for incorporating those elements, the nature and use of the 360° content is predominantly the same. While the video is playing, recipients can change the viewing angle with the mouse and thus actively participate in the action. In accordance with the explained procedure of the inertia principle of journalism

and the integration of media schemata, in which users should not be overwhelmed at the beginning of a new medium (Wolf 2014), the 360° use in the form of a photo is a widespread multimedia element and appears to be quite conclusive. Especially since "storytelling in 360 degrees presents a fascinating new creative challenge that has proved a powerful draw to commissioners and producers alike" (Rose 2018, p. 147), the same could be true for the use of VR in which the respective technology serves new design possibilities for digital long forms (Mills and Brown 2022, p. 197). However, studies show that this could lead to users being less attentive, recalling less information (Barreda-Ángeles et al. 2021, p. 154) and requiring new literacies (Rose 2018, p. 147). Nevertheless, VR enables its users to deeply interact with the virtual content (Ehrlich 2022, p. 13). This kind of participation and interactivity adds value to the overall story in the sense of vividness and authenticity and can lead to powerful reactions of the users (Ehrlich 2022, p. 11).

### 8.2. New Quality Criteria in Digital Long Forms: Gamification and Immersion

*Gamification* appeared as one new quality criterion within the analysis of the long form stories: It can consequently be described as a quality criterion that helps to encourage users from the common lean-back mode to active reception. For example, news games add value to journalism by enriching journalistic discourses through incorporating game logics and offering an engaging experience to the users (Plewe and Fürsich 2017, p. 2483). There are ethical doubts concerning news games, however, for example when it comes to sensible topics and the question of these should be displayed in games (Meier 2018, p. 429). Nevertheless, they have the chance to enhance the users' interest and empathy in the story and content (Meier 2018, p. 429).

The notion of *immersion* also appeared within the stories, its overriding component, however, being multimedia features. The most obvious in this respect are 360° elements that take the user, with or without aids, directly to the scene of the event and thus enable comprehensive insights into the event (Staschen 2017).

Through the elaborated, investigated and extended quality criteria, the audience not only has the possibility to consume the story in a reading way, but also in a visual sense on different levels. In this respect, the categories, together with their characteristics and their combinations, underline the value of the design and preparation of the content (Godulla and Wolf 2017b). The audience thus automatically becomes part of an intensive journalistic experience. The quality of a long form is thus expressed in elaborately researched and technically sophisticated content that specifically conveys background information to the audience. This development highlights the potential of digital long form stories, which can constantly surprise the audience with innovative features by combining them with multimedia and other quality criteria, and does not yet make use of all quality criteria the digital sphere offers.

### 8.3. Hypotheses for Further Investigation and Analysis

Based on the story contrasting process, several connections of elements and quality criteria were observed, which result in hypotheses that can now be tested in a more quantitative manner. First, usability was often connected with multimedia, which results in Hypothesis 1:

**H1.** *The more multimedia elements a long form story contains, the more extensive is its usability.*

Furthermore, within the *Digital Storytelling Contest*, the quality criterion of gamification only had a temporary use in the long forms. Therefore, we assume that within the canon of quality criteria, gamification plays a rather minor role, compared to aspects of multimedia or interactivity.

**H2.** *Multimedia elements and interactivity in digital long forms have a greater influence on the overall journalistic quality than playful elements.*

Finally, it became apparent that lean-back elements were frequently used, and it can be supposed that they lead to a higher immersion of the audience, which also can be tested quantitatively in future research:

**H3.** *The combination of lean-back elements contributes to a positive immersive experience for recipients of digital long forms.*

## 9. Limitations

Limitations occur regarding the novel approach of using Grounded Theory for investigating the research material at hand. Studies on assessing the quality of online content often focus on surveys among users and producers (e.g., Godulla and Wolf 2015a; McKenna et al. 2017; Radü 2019) or conducting content analysis (e.g., Riskos et al. 2019; Tulloch and Ramon 2017). In contrast, assessing quality criteria using Grounded Theory is not really common (yet).

In addition, the process of theoretical sampling could not be carried out to the end since the number of long forms was limited to 31 stories. In order to reach theoretical saturation and to be able to generalize the results, however, there should be as many long forms added until the previous theory can no longer be disproved or extended (Krotz 2018). The tendencies and subtheories identified in this study therefore serve as a basis for analyzing further stories in future research. In particular, stories with non-linear narrative structures should be integrated, whereas in this work only two of them could be identified and evaluated. In addition, future studies should measure the amount of text, video, and audio features to more accurately capture, for example, Radü's (2019) quality criteria *rhythm*.

Finally, technical limitations should be mentioned, since three long forms could no longer be played and thus it was not possible to examine the entire Interactive subcategory of the Digital Storytelling Contest of the World Press Photo Foundation.

## 10. Summary and Outlook

Summarizing the presented analysis and answering the research-guiding question, two main approaches to digital long forms could be identified within the global and professional setting of the *World Press Photo Storytelling Contest*: The linear long form story and the chapter long form story, for which our research adds new layers and nuances to previous research. Whereas both story forms were rich in continuous textual elements, videographic content and visualizations, the latter stood out with a higher degree of complexity. Therefore, more of the journalistic quality criteria were found in the chapter stories, such as transitivity and selectivity; multimedia and interactivity appeared frequently in both forms, whereas the qualities of participation and utility, as well as the use of data visualizations stay expendable.

Many of the discussed criteria, such as multimedia, participation, interactivity, and gamification play into the principles of immersion, since the "use of multiple media elements enables a better and more immersive experience of digital stories" (Radü 2019, p. 26). The results show that several audiovisual features were used in the analyzed stories to ensure such immersion. At the bottom line of this special issue, "immersive media are seen as integral to the emerging experiential market for cultural experiences". This study confirms this statement in the sense that digital stories in journalism use immersion to generate new, impressive, and enlightening experiences for their users, while keeping up with journalism's general claim to provide information and educate the public.

Answering the second part of the research-guiding question, the elaborated quality criteria were operationalized and made measurable in detail through the analysis with the help of Grounded Theory and can now be further used to obtain a comprehensive basis for the implementation and design of digital long forms. At the same time, our research provides an understanding of what has been emphasized so far in the production of long form stories regarding Internet-specific quality. Thereby, our findings add to the research concerning usability (Geis 2014; McKenna et al. 2017; Pavlic and Pavlic 2017; Pope 2020), utility (i.e., Deuze 2004; Jacobson 2012; Meier 2002; Song 2018), and multimediality (Pincus et al. 2016;

Song 2018) in digital stories. Since in this study, only award-winning stories were assessed, the findings serve both storytelling researchers as well as practitioners in various ways: For academic research, they build the basis for several further investigations concerning the quality criteria. Thereby, the assessment of quality criteria in Table 1 can be used as an orientation for investigating journalistic quality on the technical level and can be expanded to also assess the level of overall qualitative excellence. These criteria can be applied to analyze different kinds of digital stories, for example depending on the media outlet they are published in, such as by national, international, or regional media organizations. That way, a larger picture of the current state of journalistic quality in digital stories emerges. For journalistic practice, both the quality criteria as well as the detailed findings can inspire future productions. They build a knowledge base for what determinants need to be included and considered when aiming to produce a high-quality story.

Considering that journalism increasingly takes place online, and thereby faces financial struggles (Lobigs 2018; Neuberger 2002), and works under the pressure of staying connected to their audience to generate profit, media outlets need to find creative and innovative solutions to telling their stories and engage their users. Digital long forms, such as those discussed in this paper, can be one possible solution to this challenge, especially due to their immersive character and new possible modes of representation. Immersive media, such as VR, thereby have the chance to promote participation and interactivity, which raises the users' interest and can evoke powerful reactions. Nevertheless, the findings show that participation in the stories so far stays expendable. In addition, aspects of utility and the media element data visualization stay expendable as well. This might be due to higher technical competences and requirements in production as well as personnel resources in their implementation. It will be worth investigating how and if media organizations will make use of these potential benefits in the future, or if further, better applicable criteria arise.

The World Press Photo Foundation and further storytelling awards are an adequate basis for research in this field, since they consider submissions from the whole globe and marry it with professional expertise; further studies could, however, investigate long form stories as produced by single outlets or countries.

Ultimately, it can be assumed that the quality criteria will expand and develop with advancing technical possibilities—future research in the field of long form journalism and digital stories therefore must stay dynamic and sensible to technological change. While the fast-paced and changing surroundings of this research object pose a hurdle, we argue that there is no better place than in journalism research to observe this development, preserve the findings, and thereby take part in a wider discussion of a highly exciting and worthwhile phenomenon taking place globally: the further development of digital journalism and its inherent quality.

**Author Contributions:** Conceptualization, P.P.; methodology, P.P.; software, P.P.; validation, P.P., R.P. and A.G.; formal analysis, P.P.; investigation, P.P.; data curation, P.P.; writing—original draft preparation, R.P. and P.P.; writing—review and editing, R.P. and D.S.; visualization, D.S.; supervision, A.G.; project administration, R.P., A.G. and D.S.; All authors have read and agreed to the published version of the manuscript.

**Funding:** This research received no external funding.

**Institutional Review Board Statement:** Not applicable.

**Informed Consent Statement:** Not applicable.

**Data Availability Statement:** Not publicly archived data set.

**Conflicts of Interest:** The authors declare no conflict of interest.

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
