# Peer review of "Journalistic Quality Criteria under the Magnifying Glass: A Content Analysis of the Winning Stories of World Press Photo Foundation’s Digital Storytelling Contest"

_journalmedia, doi:10.3390/journalmedia3040040_

Round 1

Reviewer 1 Report

The manuscript titled Journalistic quality under the magnifying glass: A qualitative content analysis of the winning stories of World Press Photo Foundation’s Digital Storytelling Contest discusses digital long form stories and their features. The material consists of thirty-one stories, which is a rather modest number but is explained by the decision to study a particular set of stories, in this case the award-winning long forms in The World Press Photo Foundation's Digital Storytelling category, which have already been identified as high-quality stories by an independent jury.

The word "quality" is used twice in the title of the article, and it is discussed along the way. However, the concept is not defined at any point. According to the Oxford Dictionary of English quality is
1. The standard of something as measured against other things of a similar kind; the degree of excellence of something, or
2. A distinctive attribute or characteristic possessed by someone or something.
As a reader I expected to learn how the use of particular elements affects the degree of excellence (= quality in the 1st meaning) of the stories but found that the analysis focused mainly on the presence/absence of those elements (quality in the 2nd meaning) and thus remained on a rather mechanical level, which is admittedly understandable as the additional quality criteria in digital long forms (table 1) consist of technological features. There is no discussion at all about the content of the stories and how and why the recognized features such as participation or gamification actually support the storyline and bring added value to it. Thereby I suggest that the authors explain their use of the term "quality" and reconsider the title of the article, as I presume that many other readers will also interpret "qualitative content analysis" as actual content analysis instead of form analysis.

As it is, the results support earlier findings about the extensive use of continuous text in both linear and chapter stories - journalists still tend to be text-oriented. It also makes perfect sense that there was a higher number of usability elements in chapter stories than in linear stories. As such the results were as expected, and as the data was so narrow that the process of theoretical sampling could not be carried out to the end, the contribution to scholarship is limited.

What comes to the hypotheses for further investigation and analysis, I wonder if the playful elements are yet to bloom. Study has shown that journalists are rather conservative, and it is not easy for them to include new professionals in journalistic processes. Multimedia elements such as photos and videos and their creators are more familiar to journalists than coders and other tech people. In that light it seems natural that multimedia elements have found their way to digital journalism faster than for instance games or data visualizations.

Author Response

Dear Reviewer, 

thank you very much for the important advices that emerged from the reviewing process. We have adjusted the changes according to all comments and have uploaded the new version of the manuscript. We have included all reviewers' notes as comments and we have highlighted the changed text passages in yellow. 

Best regards,
The Authors 

Reviewer 2 Report

Thoughtful approach to the problem of analysis of multimedia materials. The use of WPP contest is quite useful and thorough. My only concern is the apparent disregard for analysis of videos. Unless I'm mistaken the content in the video elements was left out "the first step of the evaluation process, the long forms were compiled into documents in the form of screenshots and made available as a PDF file. This was since video 409 and animated content in particular did not meet the usual requirements for evaluation 410 during a test run. " -- 

Author Response

Dear Reviewer, 

thank you very much for the important advice that emerged from the reviewing process. We have adjusted the changes according to all comments and have uploaded the new version of the manuscript. We have included all reviewers' notes as comments and we have highlighted the changed text passages in yellow. 

Best regards,
The Authors 
